# Intra-Population Alteration on Voltinism of Asian Corn Borer in Response to Climate Warming

**DOI:** 10.3390/biology12020187

**Published:** 2023-01-26

**Authors:** Kaiqiang Liu, Zhenying Wang, Tiantao Zhang, Kanglai He

**Affiliations:** The State Key Laboratory for Biology of Plant Diseases and Insect Pests, Institute of Plant Protection, Chinese Academy of Agricultural Sciences, Beijing 100193, China

**Keywords:** voltinism, Asian corn borer, climate warming, selection, diapause

## Abstract

**Simple Summary:**

The Asian corn borer (ACB), *Ostrinia furnacalis,* is a devastating corn pest widely distributed in China. Depending on the climate, ACB may have one or up to seven generations each year, from north to south, respectively. Previous investigations suggested co-existence of the uni- and multivoltine patterns in northeast regions (45°38′ N). Efforts have been conducted to isolate uni-voltinism (with obligate diapause), multi-voltinism (with facultative diapause), and a non-diapausing strain using recurrent selections under various simulated temperature and photoperiod environments. The univoltine (Lu) strain has evolved a stable univoltinism under a diapause suppressing condition (16 h daylength at 28 °C). The multivoltine strain (Lm) was shown to have a typical facultative diapause induced by a range of short-day lengths (11–13.5 h). The majority (94.4%) of the developed Ln strain still maintained the non-diapausing nature under a diapause enhancing condition, i.e., a short (13 h) daylength at a low temperature (22 °C). The study suggests that ACB has evolutionary intra-population variation in voltinism. Under the climate change scenario warmer spring and summer will affect the proportion of sympatric voltine biotype populations that evolve toward being multivoltine.

**Abstract:**

The Asian corn borer (ACB) *Ostrinia furnacalis* (Guenée) can occur in one to seven generations annually from cool (48°00′ N) to warm (18°10′ N) region of corn cultivation in China. Although ACB is commonly known as a facultative larval diapause insect, the co-existence of various voltinism suggests that intra-population variation may have evolved for the nature of diapause, i.e., voltinism plasticity. Here, we conducted recurrent selection efforts to establish three strains of, respectively, univoltine (with obligate diapause), multivoltine (with facultative diapause), and non-diapausing ACB under various temperature and photoperiod environments. The univoltine (Lu) strain has evolved a stable univoltinism under a diapause suppressing condition (16 h daylength at 28 °C), with the diapause incidence constantly over 80% after three generations of selection. The multivoltine strain (Lm) under the high temperature (28 °C) was shown to have a typical facultative diapause induced by a range of short-day lengths (11–13.5 h). Diapause incidence was constantly <2.6% under the long day length (16 h) when the temperature was from 18 to 28 °C, i.e., low temperature could not enhance the diapause response in the Lm strain. However, the development was prolonged from 14.2 ± 0.3 d to 46.0 ± 0.8 d when the temperature was reduced from 28 °C to 18 °C. The majority (94.4%) of the developed Ln strain still maintained the non-diapausing nature under a diapause enhancing condition, i.e., a short (13 h) daylength at a low temperature (22 °C). Lm and Ln were able to complete their second generation in Heihe (50°14′ N) if the first-generation moth oviposits before 18 June. The study suggests that ACB has evolutionary intra-population variation in voltinism. Under the climate change scenario warmer spring and summer might affect the proportion of sympatric voltine biotype populations that evolve toward being multivoltine.

## 1. Introduction

To adapt to a complex environment and survive under adverse living conditions, such as low temperature in winter or lack of a host, most insects in temperate regions undergo diapause at a stage of their life history [1,2]. Although the onset of diapause, characterized by arrested development, depressed metabolism, and cold hardening, naturally occurs in various embryonic [3,4], larval [5,6], pupal [7,8], and adult [9] stages, it is usually species-specific in a single stage. In some species, such as *Charanyca trigrammica*, obligatory diapause occurs at a particular stage in every generation regardless of the environmental factors [1,10,11]. Yet, diapause is usually induced by environmental cues, including photoperiod and temperature [12,13,14], which is categorized as facultative diapause [15]. In an evolutionary context, insects living in high latitude areas have adapted diapause in their life history to enable them to survive through severe winter. Insects have also adapted voltinism plasticity to maximize their developmental and reproduction time during the favorable part of the year. Although seasonally shortening of day length mediates diapause onset, the temperature often plays a compensatory effect, i.e., elevated temperature could invert shortening day induction of diapause onset and lead to continuous development [16].

A previous study has demonstrated that climate warming has extended the plant growing season in China [17,18]. Correspondingly, earlier planting with late maturity maize variety has been adapted in cultural practice [19,20], which could offer host plant availability for adult egg laying in earlier season and larval feeding in later season. Insects can occur early in spring, and they have longer time to develop before winter. In addition, insects will develop fast under warmer climate scenario. Therefore, it is possible that insects adjust voltinism in response to the ongoing climate warming scenario. Evidence has been emerging through computer modeling with historical data [21,22,23,24,25] and empirical experiments [9].

The Asian corn borer (ACB) *Ostrinia furnacalis* (Guenée) is considered a typical facultative larval diapauser with active development during long-day photoperiods. It can have one to seven generations a year in corn-producing areas varying from north to south China [26]. Different voltine ecotypes have been reported geographically [27,28]. Several studies have been reported on the biology and ecology of the voltine ecotypes. An earlier review [27] addresses geographical race as an evolutionary breakthrough based on the fact distinct geographical populations had different voltinism under natural photoperiod conditions in the same area. Additionally, there were differences in the occurrence period, generation rhythm, and host plants in the 1960s compared to the 1980s between populations from farmland areas with corn as the main crop and forest grasslands. Latitudinal geographical populations demonstrate voltinism plasticity and variations in photoperiodic response, i.e., the lower latitudinal populations usually are trivoltine or multivoltine with shorter critical day length, whereas the higher latitudinal populations are uni- or bivoltine with longer critical day length [28,29,30,31,32,33]. 

Compensatory effects of temperature on diapause induction have been observed universally, i.e., low temperature encourages diapause directly and extends the critical daylength for diapause, whereas high temperature discourages diapause and shortens the critical daylength. [29,31]. Inheritance of diapause induction and termination are incompletely dominant, and they are influenced by interactions between the F1 genotype and photoperiod [32,33,34,35,36]. The univoltine population performs better in cold tolerance compared to the bi- and multivoltine ones [37,38,39]. The post-diapause development lasts longer in univoltine than in bivoltine ecotypes [40], which leads to better synchrony between ACB moth egg-laying and the late whorl stage (V17 stage) of corn plant development. This would result in significantly more loss than when infested at mid-whorl stage [41]. Climate warming has driven the uni- and bivoltine populations to evolve to bi- or trivoltinism [39,42,43]. Although these studies indicated that ACB has evolved geographic distinct races, strains, populations, or ecotypes, all these findings are based on the conclusion that ACB has featured voltine plasticity with facultative diapause mediated by adverse photoperiods, together with temperature interactions. Similar findings have been reported on the European corn borer (ECB) *Ostrinia nubilalis* (Hübner) in the United States [44,45,46,47,48,49].

However, it is arguable on the facultative diapause feature of ACB because investigations also show that ACB do exhibit uni- and bivoltine simultaneously in the same ecological area where the insect can usually occur for three generations and more [50]. The ACB population in Ledong, Hainan Province located in the tropical region shows a relatively low diapause incidence (6.3–6.6%) only under the conditions of 11 h or 12 h short day length and low temperature of 22 °C. The larvae do not diapause under other conditions [33]. Virtually, previous findings did not tell whether ACB evolved uni-voltinism (obligate diapause) or a non-diapausing trait within the population. If so, what is the proportion of each biotype in the population under different ecological environments and how do they respond to ongoing climate warming scenarios? Verifying these scientific issues will help to understand how phenological and/or voltinism adaptation drive biotype/ecotype variants toward trait accumulations that optimize survival and reproduction under a set of specific environmental elements. This information is useful for predicting what and how circumstances of the environment and climate conditions may affect the population dynamics. This study aimed to investigate if there is evolutionary variation in diapause strategy and how it will be affected by the key abiotic ecological factors (photoperiod and temperature) in ACB.

## 2. Materials and Methods

### 2.1. ACB Populations

Two populations of ACB, the Heihe (HH) and Harbin (L), were used in this study. The HH population was initially obtained from a field collection at Heihe (50°14′ N, 127°14′ E), Heilongjiang Province, at the end of September 2017. A total of 1803 diapause fifth instar larvae were collected by dissecting corn stalks. Those larvae were then maintained at 4 °C for 3 months for diapause termination, which could enable the larvae for post diapause development to pupate in synchrony [51,52,53]. After diapause termination, the larvae were transferred to 24 well plates (Corning, each well 3.4 mL). Each well contained one larva with a piece of wet cotton ball to supply drinking water. They were then placed in an environmental chamber at 28.0 ± 0.5 °C, 60–70% relative humidity, and a photoperiod of LD 16:8 h. Cotton balls were sprayed with water once every 3–5 days to ensure sufficient water sources available until the larva developed into a pupa. Pupae were collected every day and transferred to the chamber with the same environment for diapause termination for emergence. After eclosion, the moths were placed in an oviposition cage, as described by Zhou et al. [54]. A piece of wax paper was placed on the top of the cage as a substrate for egg laying. Egg masses were harvested each day by collecting the wax paper, with the immediate replacement for the next day′s oviposition. The offspring were used for the selection of the univoltine strain.

The L strain was originally collected in 2017 within the grass areas (action sites for moths) near corn fields in Harbin (45°38′ N, 126°34′ E), Heilongjiang Province, and had been maintained at 28 °C, 16-h day length with 60–70% relative humidity for 28 and 34 generations [39] when it was used in the experiments for selecting non-diapausing and multivoltine traits, respectively.

### 2.2. Selection for Univoltine, Multivoltine, and Non-Diapausing Strains

A modified recurrent selection approach [55] was used to establish a univoltine strain (Lu) with obligatory diapause (Figure A1, Table A1). Neonates from the HH overwintering population were reared on a meridic diet [26]. A diapause averting photoperiod of LD 16:8 h was set up throughout the selections while the temperature was increased gradually from 26 to 28 °C. Under the selection regimens, non-diapausing individuals developed to pupae in 15–20 days and proceeded development to adult emergence in another 7–10 days. Because it is visually impossible to distinguish if mature larvae are diapausing or non-diapausing, we used pre-pupal development as a criterion. Diapause event was confirmed when an individual larva reared in the experimental condition failed to pupate within 45 days. The number of pupae and diapause larvae during each rearing process was recorded. The diapausing larvae were then maintained at 4 °C for one to three months for breaking diapause and used continually for the next generation of selection; six generations of selection were proceeded. With this selecting regimen, the facultative diapause and non-diapausing traits would be eliminated.

A mass selection technique was used to develop a multivoltine strain (Lm) (Figure A2). Larvae pupated within 15 days under a diapause averting condition (16 h daylength at 28 ± 0.5 °C) were selected in each generation. The L population was initially selected for 34 generations using the method [39]. To further strengthen and confirm the multivoltinism, another 15 generations were selected in this study under the same condition, with the diapause incidence assessed in every five generations. With this selection regimen, the obligatory diapause trait would be virtually eliminated.

Part of egg masses derived from the L strain at generation 28 of selection for multivoltinism [39] was used initially to select for a non-diapausing strain (Ln) (Figure A2). A diapause-inducing photoperiod of LD 13:11 h that induced the highest incidence of diapause was set up throughout the selection, with the temperature decreased from 28.0 to 24.0 °C after 2 generations of selection. There were 8 generations of selection that proceeded in total. The number of pupae and diapausing larvae for each selecting generation were recorded, respectively. With this selecting regimen, the non-diapausing trait was selected.

### 2.3. Photoperiodic and Thermal Response

The photoperiodic responses for diapause induction in Lm and Ln strains were investigated experimentally under various photoperiods at constant temperatures of 28.0 ± 0.5 °C (Lm strain at generation 18) and 24.0 ± 0.5 °C (Ln at generation 5). Effects of temperature on diapause induction were examined under a diapause averting photoperiod of LD 16:8 h at different constant temperatures of 28.0, 26.0, 24.0, 22.0, 20.0, and 18.0 °C for the Lm strain and a diapause inducing photoperiod of LD 13:11 h at constant temperatures of 22, 24 and 30 °C for the Ln strain. There were 220–240 individuals in each of the three replicates for all the treatments. The diapause incidence and developmental duration were recorded for each treatment. All experiments were carried out in environmental chambers (RXZ-500D-LED, Jiangnan Company, Ningbo, China) with 70% RH.

### 2.4. Calculation of Effective Accumulated Temperature 

Historical monthly mean temperature data from 1970 to 2009 for Heihe and Harbin during the growing season (May to September) were obtained from the website (http://xmy.wheata.cn/, accessed on 4 July 2022), which were used to calculate each location′s monthly mean temperatures over a 10-year period (Figure A2). Daily maximum and minimum temperature data for Heihe and Harbin, from 2010–2019, were obtained from the website (http://tianqi.2345.com/wea_history/, accessed on 20 August 2020), which were used to calculate each location’s mean daily maximum and minimum temperatures over the 10-yr periods as a surrogate for recent climatic trends. Differences (δ) between the 10s (2010–2019) and 70s (1970–1979) and/or 1980s, 1990s, and 2000s of 10-yr average monthly mean temperatures were used as an increment to modify each day’s maximum or minimum temperature by subtracting δ. The effective accumulated temperature was calculated by a degree-day calculator with the method of single/horizontal developed by the University of California (http://ipm.ucanr.edu/WEATHER/index.html, accessed on 20 August 2020). The thermal threshold (baseline) for development, and degree-days for ACB life stages, were for the egg stage at 13.2 °C and 44.7 days, for the larval stage at 6.4 °C and 452.1 days, and for the pupal stage at 11.8 °C and 115.5 days [56], as well as adult preoviposition time 10.0 °C and 22.5 days [57].

### 2.5. Data Analysis

Incidence of diapause in Lu, Lm, and Ln strains under different photoperiods and temperatures, as well as larval development times of Lm strain under different temperatures, were analyzed by one-way ANOVA. Treatment means were compared using ’Fishers’ Protected LSD test to determine significant difference at a 95% confidence level. All percentage data were subjected to standard transformations to improve their normality and the homogeneity of variance. For a given dataset, if all data were within the range of 30 to 70%, no transformation was performed; if all data were within the range of either 0 to 30% or 70 to 100%, but not both, the square root was used. If all data did not follow the ranges specified here before, the arc sine transformation was used. We modeled larval development time, t(T), and over effective accumulative temperatures, *T*, using a modified reverse Gompertz model:
$$t\left(T\right)={\left\{P\times \exp\left[-\exp\left(\frac{R_{m}\times \exp\left(1\right)}{P}\right)\left(\lambda -T\right)+1\right]\right\}}^{-1}$$
in which *P* represents the minimum development time when *T* approaches infinity (maximum of suitable temperature for larval growth), while λ and Rm represent the lag and the maximum growth rate, respectively. Nonlinear convergence used SPSS nonlinear regression. In addition, data from the experiment of selection efficacy for the non-diapausing trait under recurrent selection regime were subjected to the logistic model:
$$P\left(n\right)={\left(1+\exp\left(-rn+b\right)\right)}^{-1}$$
in which *r* and *b* are the rate of increase and lag, respectively. Nonlinear convergence was using SPSS nonlinear regression. All statistical calculation processes were realized by SPSS v16.0 for Windows (SPSS Inc, Chicago, IL, USA).

## 3. Results

### 3.1. Variation of Voltinism in HH Population

After diapause termination, the time of larval development to pupate was from eight to 71 days in the HH population. When offspring were reared under the diapause averting photoperiod of LD 16:8 h at a temperature of 26 °C, there were 54.0 to 92.9% of individuals entering diapause (Figure 1). The average diapause incidence was 72.6%.

### 3.2. Development of Univoltine Traits under Selection

With recurrent selection, the incidence of diapause varied significantly across generations (*F*_8,103_ = 42.57, *p* < 0.0001) (Figure 2). Although the incidence of diapause declined from 73.6% (generation 0) to 14.7% (generation 3) along with the temperature increasing from 26 to 30 °C, it did not vary significantly among generation 3, 4, and 6 at 28 °C as well as generation 5 at 30 °C. The selection regimen (long-day photoperiod) could efficiently accumulate univoltine trait along with the increase in selection pressure (temperature). After three generations of selection, the univoltine individuals in the selected strain (Lu) were as high as 84.8%.

### 3.3. Multivoltine Strain Lm and Its Response to the Photo and Thermal Induction

Across 15 generations of selection, the diapause incidence was constantly between 0.3 and 0.5%. Therefore, it was defined as multivoltine strain Lm.

Diapause incidence of such multivoltine strain was significantly different (*F*_5,12_ = 36.57, *p* < 0.0001) in Lm larvae subjected to various regular natural photoperiods at a constant temperature of 28 °C (Figure 3). The diapause of Lm larvae was encouraged when the daylength was reduced to a range of 10–13.5 h from the long (16 h) daylength. The diapause incidence percentage (%) was the highest under the photoperiod of LD 13:11 h.

Under 16-h day length, the incidence of diapause in Lm strain varied among temperature treatments. The highest incidence of diapause was observed at the temperature of 18 °C (Figure 4), while it decreased significantly when temperature was increased from 18 to 28 °C (*F*_5,12_ = 4.86, *p* = 0.0116).

The larval developmental time of the Lm strain ranged from 14.2 to 46.0 days when it was reared at the treatment temperatures of 18 to 28 °C. There were significant differences among temperature treatments (*F*_5,12_ = 890.6, *p* < 0.0001) (Figure 4). The larval development time was reduced drastically from 18 to 22 °C and then slightly from 22 to 28 °C. The correlation between larval development time and the temperature was well fitted with a modified inverse “Gompertz Model”
$$t\left(T\right)=\left(12.537\pm 2.453\right)\exp\left\{\exp\left[\exp\left(\left(\frac{\left(1.290\pm 0.673\right)*2.7183}{12.537\pm 2.453}\right)*\left(\left(15.378\pm 0.526\right)-T\right)+1\right)\right]\right\}$$
*R*^2^ = 0.980*t*(*T*) is development time (day) and *T* is the temperature (°C).

The model showed that larva growth and development were accelerated with the increase temperatures from 18 to 22 °C, but remained steady at temperatures from 22 to 28 °C.

### 3.4. Non-Diapausing Strain Ln and Its Response to the Photo and Thermal Induction

The non-diapausing individuals in the Ln strain was rapidly accumulated using a recurrent selection approach under diapause-inducing photoperiod of LD 13:11 h. The incidence of diapause declined from 98.0 to 70.7 and 59.9 after one and two generations of selection at 28 °C. In addition, the incidence of diapause significantly declined from 97.5 to 4.8% at 24 °C after five generations of selection (*F*_5,16_ = 1199.02, *p* < 0.0001) (Figure 5). The correlation between the proportion of diapause (*P*(*n*)) and generations of selection (*n*) was well fitted with a logistic model.
$$P\left(n\right)={\left\{1+\exp\left[\left(1.556\pm 0.162\right)n+\left(-4.601\pm 0.493\right)\right]\right\}}^{-1}$$
R^2^ = 0.993.

**Figure 5 biology-12-00187-f005:**
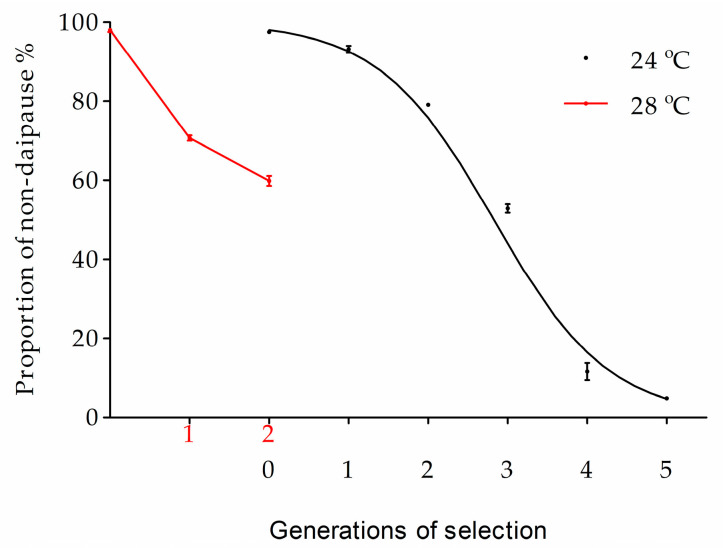
Development of the non-diapausing strain using recurrent selection. Symbols and bars represent the mean ± SE values.

Under a diapause-inducing photoperiod of LD 13:11 h, the level of incidence of diapause in the Ln strain under 24 °C was not significantly increase when larvae were subjected to low temperature of 22 °C (diapause enhancing temperature), whereas the incidence of diapause did significantly decrease when larvae were subjected at higher temperature of 30 °C (*F*_2,6_ = 113.05, *p* < 0.0001) (Figure 6).

At a constant temperature of 24 °C, changes in photoperiod significantly affected the diapause incidence of the Ln strain (*F*_4,10_ = 1068.17, *p* < 0.0001) (Figure 7). The incidence of diapause significantly decreased toward 0 when day length increased from 13 h to 14 and 16 h, whereas it significantly increased to 56.1% and 37.9% when day length decreased from 13 h to 12 h and 11 h, respectively.

### 3.5. Voltinism under Climate Warming

Based on accumulated degree-day requirements, we estimated the calendar dates of 1st and 2nd generation events for each of five 10-year periods in Heihe and Harbin (Table 1, Figure A1). From the 1970s to the 1990s, ACB could have only a single generation life cycle, and the second-generation larvae (it was partial) could not mature and enter diapause before the onset of the harsh winter in Heihe. However, a second generation emerged in the 2010s if the 1st generation moths laid eggs before 18 June, although it was a relatively small proportion of the population. 

## 4. Discussion

Through recurrent/mass selection, three voltinism distinct strains were established: (1) the univoltine strain Lu, which established a stable univoltinism under a diapause suppressing condition (16-h daylength at 28 °C), with the diapause incidence constantly over 80% after 3 generations of selection.; (2) the multivoltine strain Lm, which typically exhibited a larval diapause in response to a short day (≤11 h) photoperiod with temperature-compensation; (3) the non-diapausing strain Ln, which barely entered diapause even under a short daylength (13 h) at a temperature of 22 °C. 

Both uni- and bi-/multivoltine ecotypes in the ACB and ECB demonstrate sigmoidal photoperiod-diapause response curves, i.e., a typically long day response of insects that exhibits continuous development in summer [30,32,52]. The two ecotypes of ACB adopted distinct photoperiodic responses of diapause induction: overall the critical day length is longer in the univoltine ecotype, while the photoperiodic effect is stronger in the bi-/multivoltine ecotype, which has the property of critical daylength increase with latitude [33]. In contrast, the univoltine and/or northern ecotypes in ECB are more sensitive to the photoperiod effect [46,49,52]. This research is consistent with these findings. Under a diapause averting photoperiod of LD 16:8 h [30], there were 76.4% of individuals demonstrating univoltinism in the HH population at a constant temperature of 26 °C. Through recurrent selection using higher temperatures of 27 and 28 °C in succession, the percentage of univoltine individuals was >79%. When the selected strain Lu was subjected to an extremely high temperature of 30 °C, there were still 14.7% of individuals that were univoltine. The findings suggested that the univoltine ecotype established a stable univoltinism under a diapause suppressing condition (16-h daylength at 28 °C). In the field, the onset of diapause could be detected in every generation [27,38,39,50,58], indicating natural populations are sympatric (mixed) voltine ecotypes. In the studies on photoperiodic response in different voltine ecotypes/geographic strains, insects were collected from fields and reared under long-day photoperiods of LD 16:8 h at constant temperature (usually 26–28 °C) for one generation or more [30,32,33,34,36]. This illustrated that the insects used for experiments had been selected, at least, in one generation for multivoltine. This study used egg masses directly from field-collected overwintering larvae (HH) in the first generation. Furthermore, selection for univoltine types had been carried out in subsequent generations using the recurrent selection technique. This approach had effectively driven the HH population to evolve toward homozygous univoltine strain (Lu).

An apparent homozygous univoltine strain has been reported in the ECB [59], in contrast to our findings. Although only a single temperature (80 °F/ 26.7 °C) was used, the involvement of photoperiod in diapause determination was not observed in the study. The selection for univoltinism in our study had been conducted successively under diapause averting photoperiod of LD 16:8 h at 26, 27, and 28 °C for generations, respectively, and the univoltine trait was still heterozygous, although multivoltine individuals were few. The thermal responsiveness in selected strain Lu was mostly univoltine. Because of the long-life cycle of univoltine strain (six months or more per generation), we were unable to do further selection at the higher temperature. It is uncertain in ACB if there is a genotypic homogeneity of the univoltine ecotype.

In this study, efforts failed to establish a homozygous, non-diapausing strain through the mass selection of continuous development individuals from the L population under diapause averting photoperiod of LD 16:8 h at a temperature of 28 °C after totally 49 generations of selection. This follows the research of Gong et al. [29] using a Qingxu population (37.61° N) and a Qinshui population (35.66° N) under diapause an averting photoperiod of LD 16:8 h at a temperature of 29 °C for 36 and 44 generations of selection. These findings suggest that phenotypic univoltinism is genetically involved with recessive genes or quantitative trait loci (QTLs) in ACB. Similar findings have been reported in the ECB at a temperature of 80 °F, but photoperiod involvement in diapause was not determined [59]. 

The temperature-compensatory effect was observed in the Lm strain even under the diapause averting photoperiod of LD 16:8 h, although the incidence of diapause significantly increased only at low temperatures of 18 and 20 °C. This agrees with findings on ACB and ECB [28,29,30,31,36,60]. Additionally, our results showed that Lm larva grew and developed rapidly at a temperature ranging from 18 to 22 °C, but were steady from 22 to 28 °C, suggesting that climate warming might speed up multivoltine ecotype larval growth and development and increase voltinism further in the cold temperate region of Northeast China.

Using recurrent selection, an almost non-diapausing strain Ln (4.8% diapause under 13-h day length at 24 °C) was successively established after seven generations of selection. Similar non-diapausing dominance have been reported in the studies with the ECB [55] and a spider mite *Tetranychus urticae* (Acari: Tetranychidae) [61]. In addition, the photoperiods of LD 12:12 and 11:13 induced 56.1% and 37.9% of diapause onset, respectively, suggesting the multivoltine form had a complex genetic variability.

In northeastern China, temperatures during the growing season are relatively low, especially in Heihe where the average monthly mean temperature is between 18 to 22 °C. In addition, the increments of 10-yr average monthly temperatures range from 0.7 to 1.2 °C from May to September for the 1970s and 2010s, respectively (Figure A1), which likely resulted in accelerated larval development.

According to historical and recent surface temperature data, we predict that the first-generation moths emerged from the overwintering larvae might oviposit as early as 21 June in the 1970s and 16 June in the 2010s in Heihe, respectively. A small proportion of second-generation larvae could enter diapause before the onset of winter in the 2010s, but they could not in earlier decades. In the field, the overwintering generation ACB moth was first found on 16 June 2012, 29 June 2013, and 11 June 2014, in Nenjiang County, Heihe [62], which were consistent with our data. The findings suggest that elevated temperature has progressively driven ACB evolutionary toward multivoltine under the ongoing climate warming scenario in Heihe.

## 5. Conclusions

In conclusion, ACB has evolved univoltine, multivoltine, and non-diapausing biotypes/ecotypes, which are commonly sympatric. Univoltine and non-diapausing biotypes are likely photoperiodic non-responsive, i.e., a 16 h long day length is not effective to terminate diapause of the Lu strain under 28 °C, whereas 13-h day length is less likely to trigger diapause onset in non-diapausing biotype at 24 °C (Figure 7). However, for the multivoltine ecotype, diapause response under the long-day photoperiod could be facilitated by low temperatures. In addition, a slight increase in temperature would speed up growth and development in multivoltine biotypes at a low (18–22 °C) temperature regimen. Therefore, the ongoing climate warming scenario will drive multivoltine biotypes for increased voltinism and result in a proportion of sympatric voltine biotypes evolving toward multivoltine.

## Figures and Tables

**Figure 1 biology-12-00187-f001:**
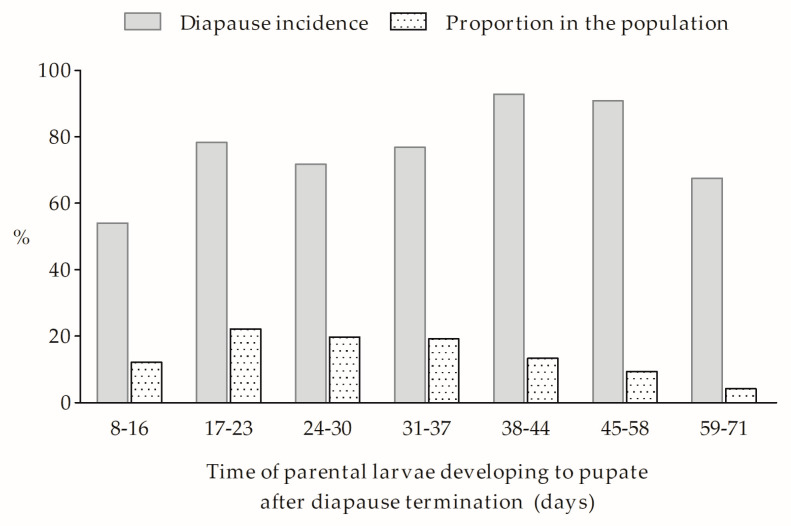
Variations in diapause and voltinism derived from the HH population of ACB. The parental larvae population is grouped according to the variation in larval development (x-axis; in days) from diapause termination to pupation. The diapause incidence rate (gray bars) and the proportion of populational univoltism (bars with dots) in offspring were summarized for each parental group.

**Figure 2 biology-12-00187-f002:**
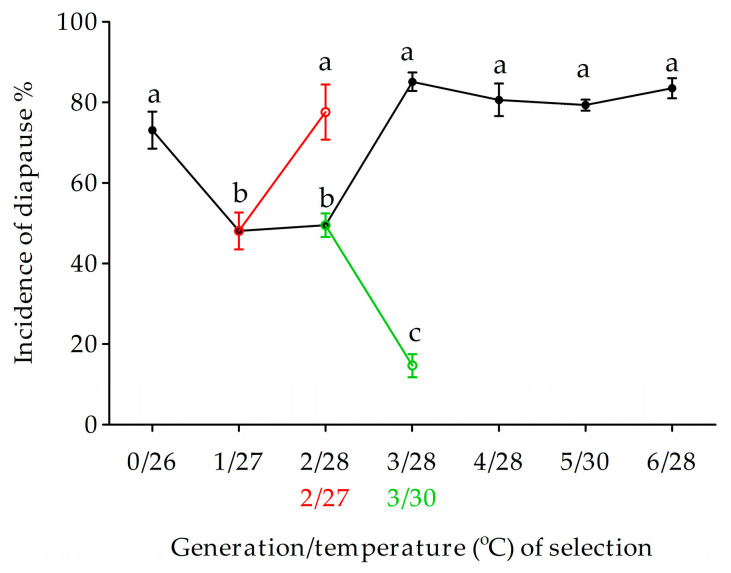
Changes in diapause incidences (%) in the process of establishing the univoltine strain (Lu) by recurrent selection under a 16 h daylength. Symbols and bars represent the mean ± SE values. Different lowercase letters indicate significant differences (*p* < 0.05) among diapause incidence of different generation/temperature selections.

**Figure 3 biology-12-00187-f003:**
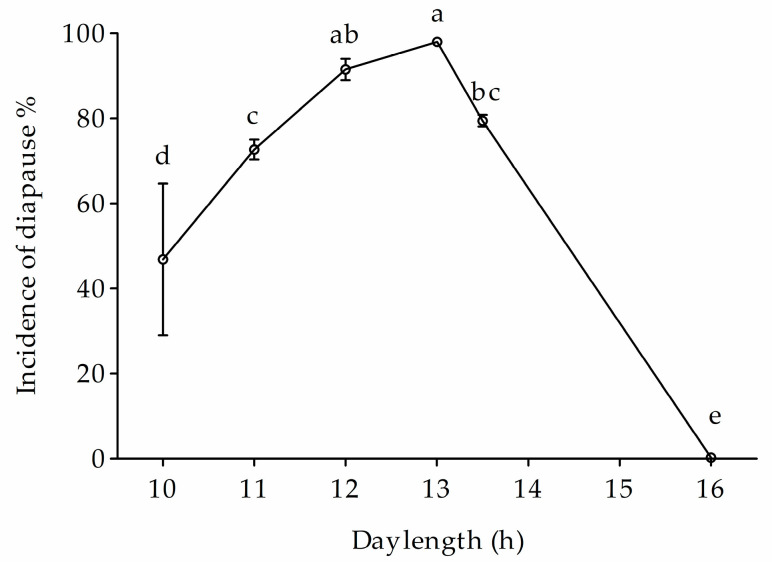
Photoperiodic response of Lm under a diapause inducing temperature (28 °C). Symbols and bars represent the mean ± SE values. Different uppercase letters indicate significant differences (*p* < 0.05) among diapause incidence of the photoperiodic response of Lm strain.

**Figure 4 biology-12-00187-f004:**
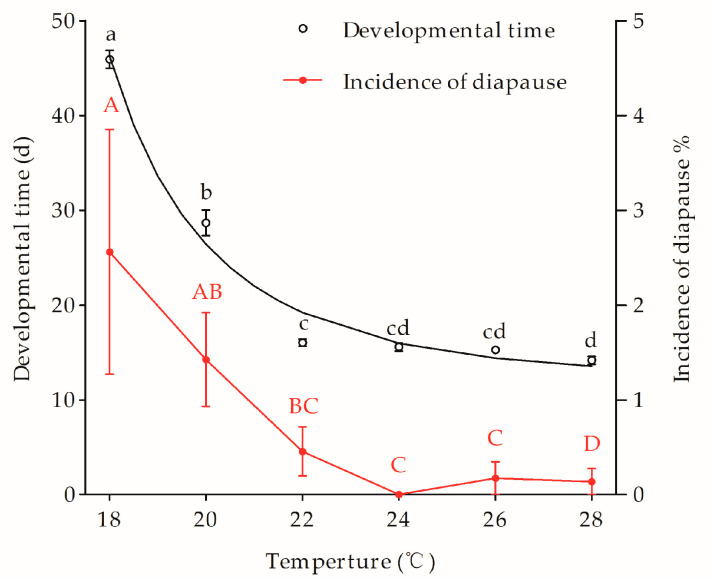
Thermal effects on larval development (days) and diapause incidence (%) in the Lm strain under a 16-h daylength. Symbols and bars represent the mean ± SE values. Different uppercase and lowercase letters indicate significant differences (*p* < 0.05) among diapause incidence in Ln strain and developmental time of Lm strain.

**Figure 6 biology-12-00187-f006:**
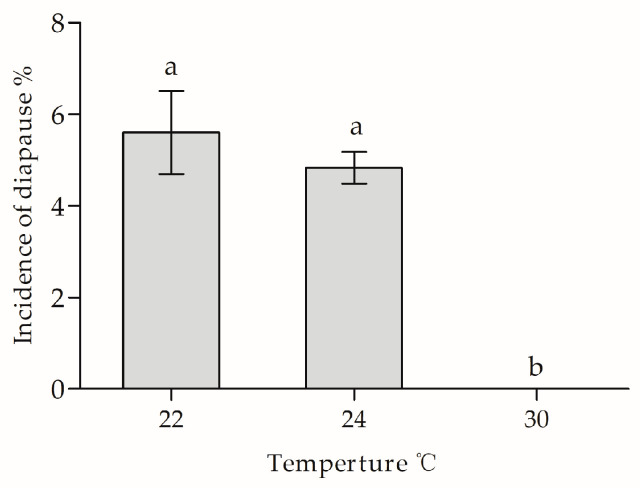
Incidence of diapause in Ln strain under different temperatures. Symbols and bars represent the mean ± SE values. Different lowercase letters indicate significant differences (*p* < 0.05) among diapause incidence in the Ln strain.

**Figure 7 biology-12-00187-f007:**
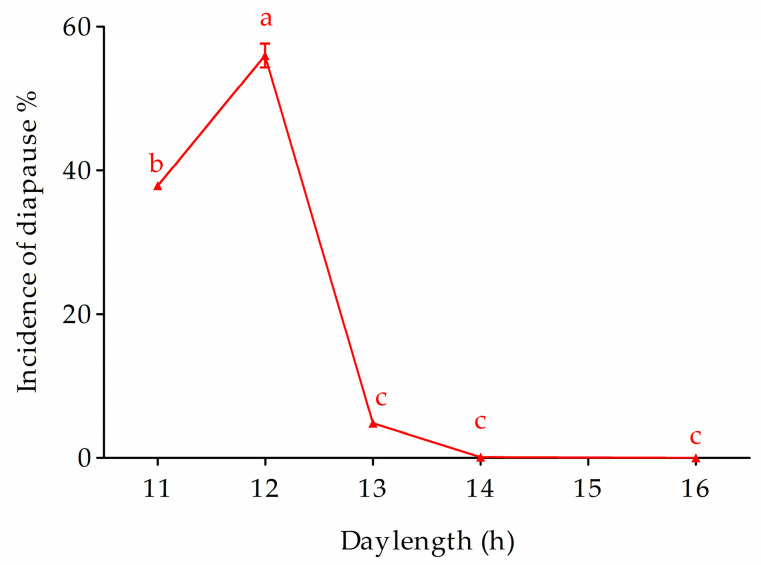
Photoperiodic response of Ln under a diapause inducing temperature (24 °C). Symbols and bars represent the mean ± SE values. Different lowercase letters indicate significant differences (*p* < 0.05) among diapause incidence of the photoperiodic response of Ln strain.

**Table 1 biology-12-00187-t001:** Predicted dates of generational events during five 10-yr periods in Heihe and Harbin.

Years	The Earliest Date of 1st Generation Egg-Laying	The Latest Date of 1st Generation Laid Eggs That Could Regenerate a Second Generation	The Earliest Date of 2nd Generation Egg-Laying	The Latest Date of 2nd Generation Laid Eggs That Could Allow Larva Develop to Mature and Enter Diapause before Winter
Heihe (50°14′ N, 127°14′ E)				
1970s	21 Jun	-	-	-
1980s	21 Jun	-	-	-
1990s	19 Jun	-	-	-
2000s	17 Jun	-	-	-
2010s	16 Jun	18 Jun	6 Aug	6 Aug
Harbin (45°38′ N, 126°34′ E)				
1970s	9 Jun	28 Jun	28 Jul	12 Aug
1980s	9 Jun	29 Jun	28 Jul	13 Aug
1990s	8Jun	2 Jul	26 Jul	13 Aug
2000s	4 Jun	6 Jul	21 Jul	19 Aug
2010s	5 Jun	7 Jul	22 Jul	18 Aug

Note: “-”means the generational event could not achieve according to the estimated accumulated degree-day.

## Data Availability

Data are contained within the article.

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
