# Peer review of "Intra-Population Alteration on Voltinism of Asian Corn Borer in Response to Climate Warming"

_biology, 2023, doi:10.3390/biology12020187_

Round 1

Reviewer 1 Report

Please find my proofreading and comments in the attached PDF. 

Author Response

Dear reviewer:

Thank you for your review comments.

Please see the attachment for details of our modification.

Reviewer 2 Report

Generally, the paper has been written well. However, some technical, and especially linguistical and methodological improvements must be made in order to increase the overall quality of presented manuscript.

Majors

1. The first and the foremost ethical issue is detected (auto)plagiarism e.g. the Figure 3 is almost the same as Figure 7 as in paper published from the same group of authors in Insects in 2021 https://doi.org/10.3390/insects12030232!

2. I strongly suggest to prepare a graphical illustration of selection methodology for the ACB voltinism detection and testing in different populations, since it will substantially improve the overall quality of the manuscript.

Minors

As for technical issues, the most obvious is writing of number ranges - it is inconsistent throughout the manuscript. Note, use only en dash (not hyphen or em dash) with no spaces either side. Use all the digits (so no elision) if at least one number in the range is from 1 to 100, e.g. 2–10, 67–69. 

Please, improve this small technical issue.

Moreover, please, write all names of species in Latin in Italic only!

Note, Ostrinia nubilalis (Hübner), so correct it in the line 102, since Guenee is not the author who revised taxonomical status of this species.

Other, detailed comments are submitted in the PDF document.

Author Response

(The authors gave the same response as above.)

Reviewer 3 Report

In this manuscript, the authors focused on the Asian corn borer and established univoltine, multivoltine, and non-diapausing strains. Their fundamental descriptions seems important for further research.

However, the importance of their experiment is quite vague.I think the former part of Introduction section is very well-written but you should rewrite the latter part of Introduction. Especially, you should explain the purpose of your experiments in detail and why you established Lu/Lm/Ln strains? Also, why you compared Lu/Lm/Ln strains should be explained in Introduction.

Minor points

Line55: "[" -> should be deleted

Line100: What is "it"?

Line279: You should add citation(s) to explain why Gomppertz model is suitable for this correlations.

Fig1: Aren't there any error bars? How many mothers were used in each category? You should add the detailed experimental design.

Fig6: Very difficult to see. Color difference is so small that you should use dotted line for purple one for example.

Author Response

(The authors gave the same response as above.)

Reviewer 4 Report

In this manuscript, Liu et al. reported the intrapopulation variance in voltinism of Ostrinia furnacalis in response to climate change. The results of this study revealed that the climate warming affects the proportion of sympatric voltine biotype populations of O. furnacalis - univoltine populations may evolve towards being multivoltine. The manuscript is clear and pleasant to read. The experiments are well designed and carried out and the results are clearly presented. Therefore, my opinion is that this manuscript may be published in Biology as it is.

Author Response

Dear reviewer:

Thank you for your review comments.

Round 2

Reviewer 1 Report

Dear authors:

The manuscript is improved substantially by your endeavor and most of my previous concerns are addressed.

Some points:

L141: has -> had

L160-165: Can be more precise if decide to remove old figure 3 to avoid plagiarism. Please see my suggestion in the PDF.

L170: please check the number “LD 13:11” here and “LD 16:11” in Figure A2.

L175: selected

L220: respectively(.)  

L250: you can add “under a 16h daylength” after “selection”

L256:Diapause incidence of such multivoltine strain”

L306: remove “incidence” after “diapause”

L310-L311: “changes in photoperiod affected significantly the diapause incidence of the Ln strain”

L343: “The two ecotypes of ACB”

L346: “latitude [33].”

L368. “study.”

L372: univoltimism -> univoltine ???? (Please check the sentence)

L374: genotypically homogeneous -> genotypic homogeneity ????

L420-421: please see the PDF for suggestion.

L565: Photoperiod ????

Author Response

(The authors gave the same response as above.)
